# Adversarial Attacks on Downstream Weather Forecasting Models: Application to Tropical Cyclone Trajectory Prediction

## Abstract

Deep learning–based weather forecasting (DLWF) models leverage past weather observations to generate future forecasts, supporting a wide range of downstream tasks, including tropical cyclone (TC) trajectory prediction. In this paper, we investigate their vulnerability to adversarial attacks, where subtle perturbations to the upstream weather forecasts can alter the downstream TC trajectory predictions. Although research on adversarial attacks in DLWF models has grown recently, generating perturbed upstream forecasts that reliably steer downstream output toward attacker-specified trajectories remains a challenge. First, conventional TC detection systems are opaque, non-differentiable black boxes, making standard gradient-based attacks infeasible. Second, the extreme rarity of TC events leads to severe class imbalance problem, making it difficult to develop efficient attack methods that will produce the attacker's target trajectories. Furthermore, maintaining physical consistency in adversarially generated forecasts presents another significant challenge. To overcome these limitations, we propose *Cyc-Attack*, a novel method that perturbs the upstream forecasts of DLWF models to generate adversarial trajectories. First, we pre-train a differentiable surrogate model to approximate the TC detector's output, enabling the construction of gradient-based attacks. *Cyc-Attack* also employs skewness-aware loss function with kernel dilation strategy to address the imbalance problem. Finally, a distance-based gradient weighting scheme and regularization are used to constrain the perturbations and eliminate spurious trajectories to ensure the adversarial forecasts are realistic and not easily detectable. Our experimental results show that *Cyc-Attack* achieves higher targeted TC trajectory detection rates, lower false alarm rates, and stealthier perturbations than conventional gradient-based attack methods.

## 1 Introduction

Recent progress in deep learning–based weather forecasting (DLWF) models (Gao et al., 2022; Lin et al., 2022; Lam et al., 2023; Bi et al., 2023) has yielded notable gains in predictive accuracy, often surpassing conventional numerical models. However, such models are vulnerable to adversarial attacks (Deng et al., 2025; Imgrund et al., 2025; Arif et al., 2025), in which subtle manipulation of the input data can lead to significant changes in the output forecasts at targeted locations. Specifying the desired weather conditions at the targeted locations is also non-trivial, as it requires considerable domain expertise to ensure that the manipulated forecasts remain physically realistic. Since the forecasts are typically used to guide downstream planning and decision-making, ***adversarial targets are more naturally defined at the downstream level***—e.g., as an altered tropical cyclone trajectory forecast rather than a perturbed set of meteorological variables—— thereby allowing an attacker to specify the target without the need for extensive domain knowledge. These downstream targets can then be used to learn the corresponding adversarial forecasts to be generated by the DLWF models.

This paper investigates the feasibility of constructing adversarial weather forecasts for downstream applications. Specifically, we focus on tropical cyclone (TC) trajectory prediction (Wang et al., 2022), given both its significant socio-economic impacts—since 1980, TCs have caused more than

$2.9 trillion in damages in the United States [1]—and the limited prior research in this critical domain. For example, Figure 1 illustrates a scenario in which an adversary manipulates the projected trajectory of Hurricane *Irene*, diverting it from its original path towards a region with extensive oil refinery infrastructure, with the intent of disrupting the energy market and profiting from the market instability. Such a manipulation could have left the densely populated U.S. East Coast under-prepared while causing costly planning and resource misallocation at the targeted region. This underscores the necessity of investigating the feasibility of adversarial attacks on TC trajectory predictions.

Generating realistic adversarial weather forecasts that alter the original TC trajectories is challenging due to the complex, non-linear interactions among the atmosphere, ocean, and environment. Over the years, researchers have developed various automated TC detection systems (Ullrich et al., 2021; Pérez-Alarcón et al., 2024; Yan et al., 2023) that leverage forecasted weather conditions generated by weather prediction models to detect and track movement of the storm. For instance, *TempestExtremes* (Ullrich et al., 2021) employs expert-defined rules to detect TC tracks from forecasted meteorological variables such as wind speed, mean sea level pressure (MSLP), and geopotential thickness.

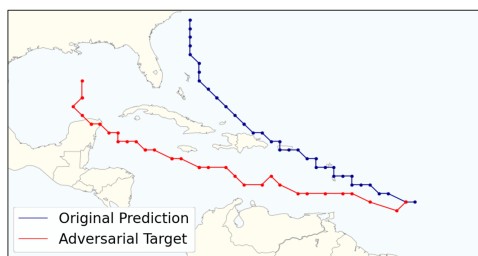

Figure 1: Adversarial manipulation of Hurricane *Irene*'s projected trajectory, generated using *TempestExtremes* software from the 10-day weather forecast of the *Graph-Cast* model, steering its original forecasted path (shown as blue line) towards a targeted region with extensive energy infrastructure (shown as red line).

While adversarial attacks on DLWF models is an active research area, their implications on downstream tasks remain largely unexplored, with several key challenges yet to be resolved. The **first** challenge is to design attack methods that can handle outputs generated by TC detection systems. Given the black-box nature of such systems, this makes gradient-based methods, such as FGSM (Goodfellow et al., 2014), PGD (Madry et al., 2017), or CW Attack (Carlini & Wagner, 2017), infeasible. One possible strategy is to pre-train a surrogate model that approximates the output of the TC detection system, enabling the use of gradient-based approaches for generating adversarial input samples. However, this strategy introduces additional challenges when applied to TC trajectory prediction. In particular, the **second** challenge is handling the extreme class imbalance problem (Johnson & Khoshgoftaar, 2019) when training the surrogate model, as the number of locations experiencing a TC event at each time step is far fewer ($< 0.01\%$) than those unaffected by the storm. Failure to address this challenge may lead to significant prediction errors when applying the TC detection system to the adversarial weather forecasts. **Third**, incorrect predictions by the surrogate model may cause gradient-based attack processes to end prematurely by falsely indicating that the target objective has been reached. The **fourth** challenge is to ensure that the adversarially generated forecasts are close to the original outputs generated by the DLWF models, while producing realistic TC trajectories that are free of conspicuous artifacts, e.g., zigzag paths, which could render the attack easily detectable.

To address these challenges, we introduce *Cyc-Attack*, a novel adversarial attack method for perturbing the forecasts of DLWF models to generate adversarial TC trajectories. First, we train a differentiable surrogate model to approximate the output of TC detection systems. Unlike gradient-free approaches, such as zeroth-order optimization (Berahas et al., 2022; Lian et al., 2016; Nesterov & Spokoiny, 2017; Chen et al., 2024), which typically require a large number of queries to the black-box, the surrogate enables the use of gradient-based approaches for efficient generation of adversarial forecasts. Next, to mitigate the severe class imbalance issue, we employ a skewness-aware loss function along with a kernel dilation strategy, guiding the surrogate model to better identify the true TC locations. The same strategy is also applied during adversarial forecast generation. Furthermore, to reduce the impact of surrogate model misprediction during adversarial forecast generation, we calibrate the false predictions by upweighting their gradient contributions to the loss, thereby ensuring the generation process does not terminate prematurely. Finally, to ensure consistency with the original DLWF forecasts and avoid unrealistic TC trajectory paths, we apply distance-based gradient weighting scheme and regularization, which constrain perturbations and suppress irrelevant trajectory distortions.

---

[1]Official public statistics from NOAA: `https://coast.noaa.gov/states/fast-facts/hurricane-costs.html`

## 2 RELATED WORK

Modern weather forecasting operates on high-dimensional geospatio–temporal data characterized by multiscale dynamics and strong cross-variable couplings. Beyond conventional numerical weather prediction (NWP) models (Côté et al., 1998; Skamarock et al., 2008; Hersbach et al., 2020) that solve the governing physical equations to produce forecasts, recent deep learning weather forecasting (DLWF) methods (Lam et al., 2023; Price et al., 2025; Gao et al., 2022; Bi et al., 2023) have emerged as promising complements. These models learn complex spatiotemporal patterns directly from data and have demonstrated improved ability to represent the atmosphere's inherently chaotic dynamics compared with conventional NWP. Those upstream forecasts are then used for downstream tropical cyclone (TC) detection and tracking. A widely used open-source tool is *TempestExtremes* (Ullrich et al., 2021), which identifies TC candidates on individual snapshots (e.g., local minima in mean sea level pressure with warm-core signatures) and stitches those satisfying specified physical criteria (e.g., wind speed is beyond a threshold) into trajectories. Other automated TC trajectory detection tools include *CyTrack* (Pérez-Alarcón et al., 2024) and *TROPHY* (Yan et al., 2023).

Adversarial attacks craft subtle perturbations to input data to manipulate a model's outputs. By attacker's knowledge (Liu et al., 2022), such methods can be classified as *white-box* (full access to model architecture and parameters), *gray-box* (partial knowledge), or *black-box* (query access only). By objective (Heinrich et al., 2024), attacks are *untargeted* (any altered outcomes), *semi-targeted* (push outputs away from a subset of outcomes), or *targeted* (force a specific desired outcome). Typical white-box methods include FGSM (Goodfellow et al., 2014), a single gradient-sign step to perturb original inputs; PGD (Madry et al., 2017), which iteratively applies the gradient-sign steps within a $\ell_\infty$-norm ball; and the Carlini–Wagner (CW) attack (Carlini & Wagner, 2017), which solves an optimization problem to find minimally perturbed adversarial examples under $\ell_0/\ell_2/\ell_\infty$ metrics. Recently, several white-box methods (Deng et al., 2025; Imgrund et al., 2025; Arif et al., 2025) have demonstrated the feasibility of attacking DLWF models. Black box attack methods generally fall into three categories. First, *surrogate/transfer* attacks train a substitute or proxy model to approximate the black-box and then craft inputs that transfer across models for achieving the same attacking performance (Papernot & McDaniel, 2016; Papernot et al., 2017). Second, *zeroth-order (ZO)* or gradient-free methods estimate descent directions from queries. Two standard estimators are coordinate-wise finite differences and randomized vector-wise schemes (e.g., NES-style). For example, *DeepZero* (Chen et al., 2024) scales ZO optimization via efficient coordinate-wise estimators, sparsity priors, and parallelism. Third, *decision-based random-search* (Brendel et al., 2018) methods operate with label-only feedback—e.g., Boundary Attack starts from an adversarial point and reduces the perturbation norm; HopSkipJump (Chen et al., 2020) follows the decision boundary using binary queries; and Square Attack (Andriushchenko et al., 2020) performs randomized, block-wise updates that are query-efficient and support targeted variants. In this work, we adopt a surrogate-model strategy for crafting our attacks.

The severe class imbalance problem motivates the design of new loss functions, including focal loss (Lin et al., 2017) that down-weights easier negative examples while focusing on harder examples; Tversky loss (Salehi et al., 2017) that tunes precision–recall trade-offs via asymmetric penalties; and IoU-surrogates such as Lovász-Softmax (Berman et al., 2018) for direct optimization of set similarity. In this paper, we employ a variation of focal loss to mitigate the class imbalance problem.

## 3 PROBLEM STATEMENT

Let $\mathcal{D} = (\mathbf{D}_1, \mathbf{D}_2, \cdots, \mathbf{D}_T)$ denote a multivariate, gridded weather dataset, where $\mathbf{D}_t \in \mathbb{R}^{d \times r \times c}$ represents the observations of $d$ weather variables on a spatial grid of dimension $r \times c$ at time $t$. At each time step $t_0$, we define the predictor $\mathbf{X}(t_0) = \{\mathbf{D}_{t_0-\alpha-1+\tau}\}_{\tau=1}^{\alpha+1}$, a window of $\alpha + 1$ past observations up to $t_0$, and the target $\mathbf{Y}(t_0) = \{\mathbf{D}_{t_0+\tau}\}_{\tau=1}^{\beta}$, a window of $\beta$ future observations. For brevity, we denote them as $\mathbf{X}$ and $\mathbf{Y}$, respectively. We consider a weather fore-

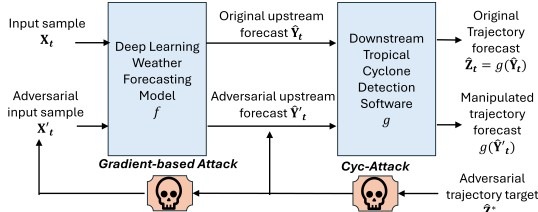

Figure 2: Adversarial attack on downstream tropical cyclone (TC) trajectory prediction.

casting model $f$ that generates upstream predictions, $\hat{\mathbf{Y}} = f(\mathbf{X})$, for the future observations $\mathbf{Y}$. Based on the upstream forecasts $\hat{\mathbf{Y}}$, a downstream black box detector $g$ will produce a TC trajectory $\hat{\mathbf{Z}} = g(\hat{\mathbf{Y}})$, where $\hat{\mathbf{Z}} \in \{0, 1\}^{\beta \times r \times c}$. Each entry $\hat{Z}_{tij} = 1$ indicates that location $(i, j)$ is experiencing a TC event at time $t$, while $\hat{Z}_{tij} = 0$ denotes otherwise. Let $\hat{\mathbf{Z}}^* \in \{0, 1\}^{\beta \times r \times c}$ be the attacker's intended adversarial trajectory. We aim to generate adversarial upstream forecasts $\hat{\mathbf{Y}}' \in \mathbb{R}^{\beta \times d \times r \times c}$ by perturbing $\hat{\mathbf{Y}}$ so that the black-box detector $g$ produces a TC trajectory $g(\hat{\mathbf{Y}}')$ that closely matches $\hat{\mathbf{Z}}^*$, thereby altering the original downstream prediction $\hat{\mathbf{Z}}$.

To enable gradient-based attacks, we approximate the black-box detector $g$ with a probabilistic, differentiable surrogate model $\tilde{g}$. Given an upstream forecast $\hat{\mathbf{Y}}$, let $\hat{\mathbf{P}} = \tilde{g}(\hat{\mathbf{Y}}) \in [0, 1]^{\beta \times r \times c}$ be the surrogate model output, where $\hat{P}_{tij}$ is the probability that location $(i, j)$ at time $t$ is experiencing a TC event. The probabilities can be converted into binary predictions by thresholding at 0.5, i.e., assigned to 1 if $\hat{P}_{tij} \geq 0.5$ and 0 otherwise. During pre-training, we seek to learn a surrogate model $\tilde{g}$ that produces TC trajectories closely approximating those of the black-box detector, i.e., $\tilde{g}(\hat{\mathbf{Y}}) \approx g(\hat{\mathbf{Y}})$. The adversarial upstream forecast $\hat{\mathbf{Y}}'$ can then be generated by applying a gradient-based attack algorithm $\mathcal{A}$ on the surrogate model output, i.e., $\hat{\mathbf{Y}}' = \mathcal{A}(\tilde{g}, \hat{\mathbf{Y}}, \hat{\mathbf{Z}}^*)$.

In this paper, we employ *GraphCast* (Lam et al., 2023), a leading DLWF model developed by Google DeepMind, as the forecasting model $f$ to generate the upstream forecasts $\hat{\mathbf{Y}}$. We then use *TempestExtremes* (Ullrich et al., 2021) as the downstream TC trajectory detector $g$ (see Appendix B for details). Nevertheless, due to the model-agnostic nature of the formulation, the proposed adversarial attack method is applicable to other DLWF models and TC trajectory detectors.

## 4 CYC-ATTACK: ADVERSARIAL ATTACK ON TC TRAJECTORY PREDICTIONS

Our *Cyc-Attack* framework comprises of 2 main components: (1) a probabilistic, differentiable surrogate model trained to approximate the TC detection system and (2) an adversarial weather forecast generation module. Given the original DLWF forecast $\hat{\mathbf{Y}}$, *Cyc-Attack* generates an adversarial upstream forecast $\hat{\mathbf{Y}}'$ that, when processed by the TC detection system, produces a TC trajectory closely aligned with the adversary's target trajectory, $\hat{\mathbf{Z}}^*$. The adversarial upstream forecast can then be fed into existing DLWF attack methods (Deng et al., 2025; Imgrund et al., 2025; Arif et al., 2025) to learn the adversarial input $\mathbf{X}'$ that yields the manipulated forecast.

### 4.1 SURROGATE MODEL PRE-TRAINING

To pre-train the surrogate model $\tilde{g}$ into approximating the black box model $g$, we consider a training data of the form $\{(\hat{\mathbf{Y}}_t, \hat{\mathbf{Z}}_t)\}$, where $\hat{\mathbf{Y}}_t \in \mathbb{R}^{d \times r \times c}$ and $\hat{\mathbf{Z}}_t \in \{0, 1\}^{r \times c}$. The data (1) aligns with the outputs of many TC detectors such as *TempestExtremes*, which first identify a set of locations affected by the TC and then stitch them to form trajectories, (2) introduces flexibility by relying solely on spatial dimensions, allowing the surrogate model to operate across different forecast horizons without the need for re-training, and (3) enables the use of image segmentation models such as *DeepLabV3+* (Chen et al., 2018) as surrogate model $\tilde{g}$, trained to fit $\hat{\mathbf{Z}}_t$ from $\hat{\mathbf{Y}}_t$, i.e., $\tilde{g}(\hat{\mathbf{Y}}_t) \approx \hat{\mathbf{Z}}_t$.

Pre-training the surrogate model requires addressing the severe imbalance issue (Johnson & Khoshgoftaar, 2019), where TC-affected locations at each time step $t$ are extremely sparse relative to non-TC locations ($< 0.01\%$). As a result, the surrogate model is biased towards the non-TC locations (i.e., majority class), leading to higher false negative rates. To mitigate this issue, the surrogate model is trained to predict dilated regions around each TC-affected location rather than predicting each location $(i, j)$ strictly as TC ($\hat{Z}_{tij} = 1$) or non-TC ($\hat{Z}_{tij} = 0$). The dilated TC region data $\hat{\mathbf{Z}}_t^D$ are generated from the original TC location data $\hat{\mathbf{Z}}_t$ using a truncated Gaussian kernel, where given $(i, j)$, we define a neighborhood $\Omega_{ij} = \{(p, q) : (p - i)^2 + (q - j)^2 \leq R^2\}$:

$$\forall (p, q) \in \Omega_{ij} : \hat{Z}_{tpq}^D = \max\left\{ \hat{Z}_{tpq}, \ \hat{Z}_{tij} K_{\sigma, R}(p - i, \ q - j) \right\}, \tag{1}$$

where $K_{\sigma, R}(u, v) = \exp\left( -\frac{u^2 + v^2}{2\sigma^2} \right)$ if $\sqrt{u^2 + v^2} \leq R$, and 0 otherwise. Here, $\sigma$ is the kernel parameter and $R$ sets the maximum dilation radius. Intuitively, this process expands the TC locations

to its nearby affected regions, producing soft labels $\hat{Z}_t^D \in [0, 1]^{r \times c}$ instead of hard $\{0, 1\}$ labels. If a data point resides within multiple neighborhoods, e.g., if it lies close to two distinct TC trajectories, we assign its dilated label by taking the minimum of the values induced by those neighborhoods.

To alleviate the class imbalanced issue, the surrogate model is trained on $\{(\hat{Y}_t, \hat{Z}_t^D)\}$ by minimizing the following skewness-aware loss, which is a variation of the focal loss (Lin et al., 2017):

$$\mathcal{L}_{\text{surrogate}} = -\frac{1}{\beta} \sum_{t=1}^{\beta} \sum_{p=1}^{r} \sum_{q=1}^{c} \left[ (1 - \hat{P}_{tpq})^2 \, \hat{Z}_{tpq}^D \log(\hat{P}_{tpq}) + (\hat{P}_{tpq})^2 (1 - \hat{Z}_{tpq}^D) \log(1 - \hat{P}_{tpq}) \right], \quad (2)$$

where $\hat{P}_{tpq}$ is the probability predicted by the surrogate $\tilde{g}$ that location $(p, q)$ at time $t$ lies within the dilated TC region with $\hat{Z}_{tpq}^D$ denotes the corresponding dilated ground-truth label. Training is performed with the Adam optimizer (Kingma, 2014).

The focal loss alone (without kernel dilation) is insufficient because it can produced false positives scattered all over the map. However, by combining it with our kernel dilation strategy, we empirically show that the scattered false positives are substantially reduced (see Table 1 and Figure 3). The remaining false positives are mostly concentrated as localized clusters around each true TC location. One explanation is that standard focal loss is commonly used in small object detection from images, where it emphasizes uncertain boundary pixels to refine localization. Without dilation, however, a TC location lacks meaningful boundary context, causing focal loss to highlight scattered noisy points instead. With dilation, local neighborhoods are restored around each TC location, enabling focal loss to capture boundary uncertainty within coherent regions rather than isolated noise.

## 4.2 ADVERSARIAL WEATHER FORECAST GENERATION

Given the trajectory $\hat{Z}$ generated from the original weather forecast $\hat{Y}$, let $\hat{Z}^*$ be the adversarial trajectory provided by an attacker. Our aim is to generate an adversarial upstream forecast $\hat{Y}_t'$ at each time step $t$ by perturbing $\hat{Y}_t$ in such a way that $g(\hat{Y}_t') \approx \hat{Z}_t^*$. Projected Gradient Descent (PGD) (Madry et al., 2017) is a standard gradient-based attack, where given a differentiable loss $L(g(\hat{Y}_t), \hat{Z}_t^*)$, it iteratively perturbs the original upstream forecast $\hat{Y}_t$ by taking the following gradient steps to minimize the loss: $\hat{Y}_t' = \hat{Y}_t - \eta \, \text{sign}(\nabla_{\hat{Y}_t} L)$ where $\eta$ is the step size. After each step, the perturbed input is clipped to ensure it stays within a small neighborhood (e.g. an $\ell_\infty$ ball of radius $\delta$) to remain close to the original forecast, i.e., $\|\hat{Y}' - \hat{Y}\|_\infty \leq \delta$. However, standard PGD attacks are insufficient for adversarial trajectory attacks due the following challenges.

**First**, we observe that using $\hat{Z}_t^*$ directly to identify the targeted locations for perturbation often causes the attack algorithm to generate adversarial upstream forecast $\hat{Y}_t'$ that, when processed by the surrogate model $\tilde{g}(\cdot)$, yields trajectories with substantial false positive locations dispersed across the map, making the attack easier to be detected. This issue parallels the class imbalance problem described in the previous section. To address this, we adopt the same kernel dilation strategy of Equation (1) to construct a dilated adversarial downstream target $\hat{Z}_t^{*D} \in [0, 1]^{r \times c}$ from $\hat{Z}_t^*$.

**Second**, any misclassification by the surrogate model, at either the original or targeted trajectory locations, will distort adversarial forecast generation and cause premature termination of the attack algorithm. To illustrate, consider the case where the attacker aims to label a targeted location as TC-affected even though the original trajectory labels it as otherwise, i.e., $\hat{Z}_{tij}^* = 1$ while $\hat{Z}_{tij} = 0$. If the surrogate incorrectly predicts the location to be TC-affected, i.e., $\tilde{g}(\hat{Y}_{tij}) = 1 \neq \hat{Z}_{tij}$, the algorithm will refrain from perturbing the forecast at that location, mistakenly assuming the target has been achieved. A similar problem arises when $\hat{Z}_{tij}^* = 1$ while $\hat{Z}_{tij} = 0$. To mitigate this issue, we introduce a calibration mask $\mathbf{M} \in \{0, 1\}^{\beta \times r \times c}$ to identify the set of locations $\{(i, j)\}$ where the surrogate's initial predictions $\tilde{g}(\hat{Y}_{tij})$ incorrectly match the adversarial target $\hat{Z}_{tij}'$ at time $t$:

$$M_{tij} = \mathbf{1}(\hat{Z}_{tij}^* \neq \hat{Z}_{tij}) \mathbf{1}(\tilde{g}(\hat{Y}_{tij}) = \hat{Z}_{tij}'). \quad (3)$$

As will be described below, the calibration mask is used to avoid premature termination, ensuring that perturbations proceed even when the surrogate's initial prediction matches the adversarial target.

**Third**, while kernel dilation reduces scattered false positive locations, it expands the target region, thereby increasing the number of candidate locations that can be stitched by the TC detection system to form a TC trajectory. This leads to TC trajectories with zigzag paths as shown in Figure 5, making them unrealistic and easily detectable. Thus, to improve stealthiness and realisticness, we constrain the perturbation on $\hat{\mathbf{Y}}$ by down-weighting the gradient update for locations far away from the original and targeted paths and up-weighting the penalty for such locations in the loss function. Specifically, let $\mathcal{S}_t = \{(i,j) : \hat{Z}^*_{tij} = 1\}$ be the set of target TC-affected locations at time $t$. For any location $(p,q)$, define its geodesic distance to nearest target location as $d_t(p,q) = \min_{(i,j) \in \mathcal{S}_t} \arccos\big(\sin\phi_p \sin\phi_i + \cos\phi_p \cos\phi_i \cos(\lambda_p - \lambda_j)\big)$ if $\mathcal{S}_t \neq \emptyset$, or 0 otherwise, where $(\phi_i, \lambda_j)$ denote the corresponding latitude-longitude of $(i,j)$. The geodesic distance to nearest target is used to compute the following two distance-based weights for each location $(p,q)$:

$$
w^{\text{grad}}_{tpq} = \begin{cases} 1, & (p,q) \in \mathcal{S}_t, \\ \exp\big(-\frac{d_t(p,q)^2}{2\sigma^2_{\text{grad}}}\big), & (p,q) \notin \mathcal{S}_t, \end{cases} \qquad w^{\text{reg}}_{tpq} = \begin{cases} 0, & (p,q) \in \mathcal{S}_t, \\ 1 - \exp\big(-\frac{d_t(p,q)^2}{2\sigma^2_{\text{reg}}}\big), & (p,q) \notin \mathcal{S}_t. \end{cases} \tag{4}
$$

We use $w^{\text{grad}}$ to modify the gradient update step of PGD as follows:

$$
\hat{Y}'^{(k+1)}_{tij} = \text{Clip}_\delta\left(\hat{Y}'^{(k)}_{tij} - \eta \cdot w^{\text{grad}}_{tij} \cdot \text{sign}\left(\frac{\partial \mathcal{L}_{\text{adv}}}{\partial \hat{Y}'^{(k)}_{tij}}\right)\right), \tag{5}
$$

where $k$ is the iteration number, $\eta$ is the step size, $\text{sign}(\cdot)$ denotes element-wise sign function and $\text{Clip}_\delta(\cdot)$ projects the perturbed input onto the $\ell_\infty$-ball of radius $\delta$ around $\hat{Y}_{tij}$. In this scheme, the target locations always receive full gradient updates ($w^{\text{grad}}_{tpq} = 1$) while non-target locations are updated with exponentially decaying weights based on their distance to nearest target. Additionally, the loss function is modified to incorporate the distance-based weight $w^{\text{reg}}_{tpq}$ as follows:

$$
\mathcal{L}_{\text{adv}} = -\frac{1}{\beta} \sum_{t=1}^{\beta} \sum_{i=1}^{r} \sum_{j=1}^{c} \big[(1 - \hat{P}'_{tij})^\gamma \hat{Z}^{*D}_{tij} \log(\hat{P}'_{tij}) + \hat{P}'^\gamma_{tij}(1 - \hat{Z}^{*D}_{tij}) \log(1 - \hat{P}'_{tij})\big]
$$
$$
+ \lambda \, \|w^{\text{reg}}_{tij}(\hat{Y}_{tij} - \hat{Y}'_{tij})\|^2_2, \tag{6}
$$

where $\hat{P}'_{tij}$ is the surrogate predicted probability for adversarial input $\hat{\mathbf{Y}}'$, $\hat{Z}^{*D}_{tij}$ is the dilated adversarial target, $\beta$ is the number of time steps, and $\lambda$ is regularization hyperparameter. The focal loss parameter $\gamma$ depends on calibration mask $\mathbf{M}$: $\gamma = 2$ if $M_{tij} = 0$ and $\gamma = 0$ if $M_{tij} = 1$. The distance-based regularization with calibration mask ensures that non-targeted locations are penalized increasingly with distance while locations near the target incurs less penalties.

## 5 EXPERIMENTAL EVALUATION

### 5.1 EXPERIMENTAL SETUP

**Datasets** We conducted our experiments using datasets from the following two sources: (1) *IB-TrACS* (International Best Track Archive for Climate Stewardship), containing the latitude and longitude coordinates of historical TC tracks at 6-hourly intervals, along with their intensity and other meteorological features. (2) *ERA5 Reanalysis Data*, which provides hourly global weather observations from 1979 to 2018 at $1° \times 1°$ spatial resolution. We constructed two datasets, *TC1* and *TC2*, from these sources. *TC1* contains 285 TCs recorded between January 1, 2019 to January 1, 2025, along with their corresponding weather data from ERA5 Renalysis. For each TC, trajectory information is retained for their first 14 time steps, with the initial 2 time steps used as input for the *GraphCast* model and the remaining 12 time steps reserved for forecasting. *TC2* is a smaller dataset consisting of 10 major TCs selected for case studies. Each TC contains location information for 40 consecutive 6-hourly time steps and their associated ERA5 Reanalysis weather data. In this case, the *GraphCast* model predicts 38 future time steps from the first 2 time steps. For both datasets, *TempestExtremes* is applied to the *GraphCast* forecasts to generate downstream TC trajectories. These trajectories are then used to construct the adversarial downstream targets. See Appendix C.1 for further details on the datasets and their pre-processing.

Table 1: Impact of dilation radius $R$ on the segmentation performance of the pre-trained *DeepLabV3+* (Xception backbone) used as the surrogate model, evaluated on a test set of 684 global 1° maps ($180 \times 360$ grid each) containing 578 TC and 41,701,822 non-TC locations.

| $R$ | TNR↑ | FNR↓ | FPR↓ | TPR↑ |
|---|---|---|---|---|
| 0 | 0.9933 | 0.0104 | 0.0067 | 0.9896 |
| 1 | 0.9999 | 0.4135 | 0.0001 | 0.5865 |
| 2 | 0.9998 | 0.1869 | 0.0002 | 0.8131 |
| 3 | 0.9993 | 0.0692 | 0.0007 | 0.9308 |
| 5 | 0.9986 | 0.0606 | 0.0014 | 0.9394 |

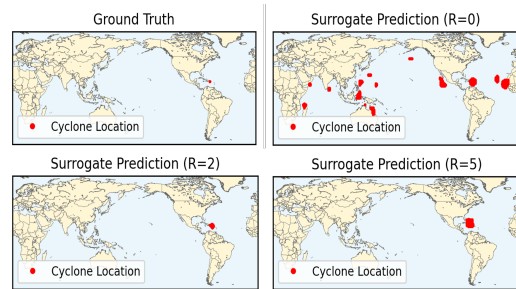

Figure 3: Visualization of TC location predictions using *DeepLabV3+* (*Xception* backbone) trained on labels produced with different dilation radii ($R \in \{0, 2, 5\}$). The case $R = 0$ corresponds to no dilation. Predictions are compared against ground-truth detections from *TempestExtremes* (top-left).

**Baseline Methods** We compared the performance of *Cyc-Attack* against the following baseline methods: *ALA* (Ruan et al., 2023), which leverages Adam-based updates to attack DLWF models; *TAAOWPF* (Heinrich et al., 2024), which leverages PGD-based updates; *AOWF* (Imgrund et al., 2025), which adopts a cosine-scaled step-size schedule to anneal update magnitudes during iterative attacks. All attackers are implemented using a pre-trained *DeepLabV3+* (Chen et al., 2018) surrogate model to mount attacks against the black-box *TempestExtreme*. For each attack, we run 1,000 iterations to ensure convergence. We use a gradient clipping threshold of $\delta = 10.0$ on standardized values to allow sufficient perturbations to create or remove TC events; and the attack step size is set to $\eta = 0.01$. More details on training and hyperparameter tuning are in Appendix C.2.

**Evaluation Metrics** We evaluated the performance of the competing methods in terms of the following criteria: (1) ***Detection accuracy at individual TC-location level*** by comparing adversarial target locations with the predicted locations obtained from adversarial upstream forecasts. The specific metrics used include true positive rate (TPR), true negative rate (TNR), false positive rate (FPR), and false negative rate (FNR). (2) ***Detection accuracy at the TC-trajectory level*** in terms of TC detection rate (DR) and TC false alarm rate (FAR). To compute these metrics, we first determine the number of overlapping locations between an adversarial target trajectory and its closest predicted trajectory derived from the adversarial upstream forecasts. Two locations are considered overlapping if their great-circle distance is less than $R = 2$. An adversarial target trajectory is deemed successfully detected if at least 50% of its location points overlap with the corresponding predicted trajectory. If it shares 0 location points, the predicted trajectory is considered a false alarm. (3) ***Closeness*** ($\delta_C$), which measures the $\ell_1$-norm between the original upstream forecast $\hat{\mathbf{Y}}$ and its adversarial counterpart $\hat{\mathbf{Y}}'$. A smaller closeness value indicates a stealthier attack.

## 5.2 Experimental Results

**Performance of the Surrogate Model** The surrogate model demonstrates strong overall skill, but its sensitivity depends on the dilation radius $R$. As shown in Table 1, no dilation $R = 0$ attains the highest recall (TPR = 0.9896) but with elevated false alarms (FPR = 0.0067) scattered throughout the map, whereas $R = 1$ sharply reduces recall (TPR = 0.5865). A moderate dilation ($R = 2$) achieves a balanced trade-off (TPR = 0.8131, FPR = 0.0002), and $R = 5$ recovers higher recall (TPR = 0.9394) with only marginal increase in false positives. The example in Figure 3 shows that kernel dilation helps to reduce scattered false positives by concentrating them into compact clusters surrounding the targeted locations. For these reasons, we adopt a pre-trained surrogate with $R = 2$ for subsequent experiments as it provides a compromise between sensitivity and specificity.

**Performance Comparison of Adversarial Attack Methods** *Cyc-Attack* attains the most favorable trade-off among evaluated methods: it achieves the highest trajectory detection rate (DR) while maintaining low false-alarm rate (FAR) and one of the smallest perturbation magnitudes $\delta_C$, together with strong location-level performance (low FPR/FNR and high TPR). All the baselines (*AOWF*,

Table 2: Comparison of different methods on TC1 and TC2 datasets in terms of their detection accuracies at location-level (FPR, FNR, TPR), trajectory-level (DR and FAR), and closeness ($\delta_C$). Cyc-Attack-$d$ and Cyc-Attack-$w$ denote variants of Cyc-Attack that omit the kernel-dilation strategy (Equation 1) and the weighting scheme (Equation 4), respectively. In the table, the best value for each metric is printed in **bold**. For all reported Cyc-Attack results, the dilation radius is set to $R = 1$.

| Method | TC1 | | | | | | TC2 | | | | | |
|---|---|---|---|---|---|---|---|---|---|---|---|---|
| | FPR↓ | FNR↓ | TPR↑ | DR↑ | FAR↓ | $\delta_C$↓ | FPR↓ | FNR↓ | TPR↑ | DR↑ | FAR↓ | $\delta_C$↓ |
| ALA | **0.0001** | 0.8218 | 0.1782 | 0.3196 | 0.1393 | 0.0459 | 0.0007 | 0.7380 | 0.2620 | 0.2222 | 0.5925 | 0.0466 |
| TAAOWPF | 0.0002 | 0.8640 | 0.1360 | 0.1037 | 0.5990 | 0.1752 | 0.0045 | 0.7812 | 0.2188 | 0.0211 | 0.8802 | 0.1948 |
| AOWF | **0.0001** | 0.7971 | 0.2029 | 0.2977 | 0.1526 | 0.0449 | 0.0012 | 0.7067 | 0.2933 | 0.1000 | 0.7000 | 0.0499 |
| Cyc-Attack | **0.0001** | **0.7001** | **0.2999** | **0.5570** | **0.0536** | **0.0003** | **0.0004** | **0.4447** | **0.5553** | **0.7500** | **0.0833** | **0.0002** |
| Cyc-Attack-$d$ | **0.0001** | 0.8476 | 0.1524 | 0.2803 | 0.0934 | **0.0003** | **0.0004** | 0.5962 | 0.4038 | 0.5000 | 0.2083 | **0.0002** |
| Cyc-Attack-$w$ | 0.0002 | 0.8640 | 0.1360 | 0.1037 | 0.5990 | 0.1752 | 0.0006 | 0.6971 | 0.3029 | 0.0274 | 0.8901 | 0.2168 |

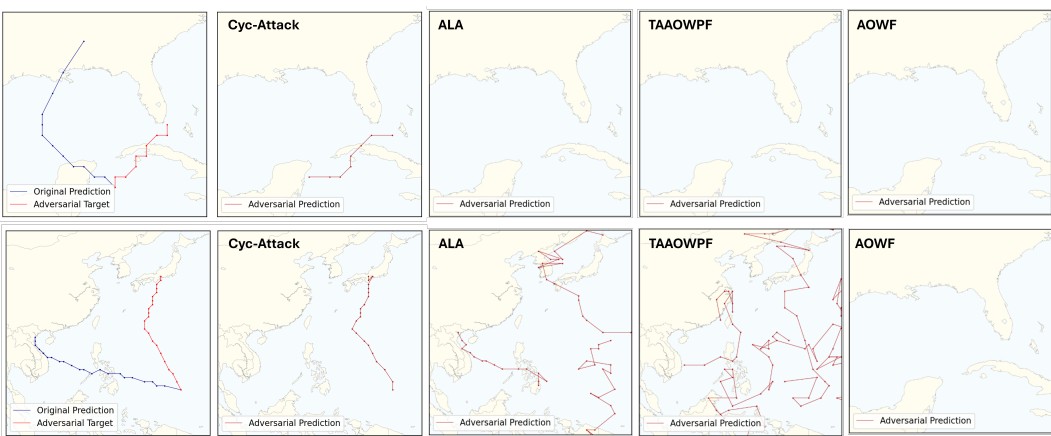

Figure 4: Visualization of adversarial attacks. Top: Hurricane *Delta* (from 10/26/2020 to 11/05/2020); Bottom: Typhoon *Haiyan* (from 11/03/2013 to 11/13/2013). For each case, the left panel shows the original trajectory (blue) detected by *TempestExtremes* from upstream *GraphCast* forecasts and the adversarial target trajectory (red); the right panels show the produced by different baseline methods using the pretrained surrogate model.

*ALA, TAAOWPF*) underperformed relative to *Cyc-Attack*. They attain lower DR while incurring larger $\delta_C$ or producing fragmented, high-FAR detections (as evidenced in Table 2 and Figure 4). The superior performance of *Cyc-Attack* can be explained as follows: the distance-weighting in *Cyc-Attack* concentrates perturbation strength near the adversarial target, which suppresses off-target perturbations and reduces false-positive trajectories, while kernel dilation focuses the loss and gradients within an expanded target neighborhood, improving spatial coherence of the induced forecast perturbation. Existing gradient-based attack methods lack these localized, distance-sensitive mechanisms and therefore either spread perturbations broadly (yielding false positives) or produce dispersed signals that background variability can deflect, preventing reliable alignment with the intended target trajectory. For ablation study, we considered 2 variations of our method——*Cyc-Attack-d* (no dilation) and *Cyc-Attack-w* (no distance weighting). The results suggest a lower DR and/or higher FAR and larger $\delta_C$, which verifies the importance of incorporating kernel dilation and distance weighting to generate more realistic and accurate adversarial target trajectories.

**Detailed Analysis** (1) We examine the sensitivity of *Cyc-Attack* to dilation radius $R$ (Figure 5) and observe that larger radii produce more zigzags in adversarial trajectories, as the perturbation region extends beyond the target. (2) We apply PCA-based anomaly detection (Abdi & Williams, 2010), Isolation Forest (IF) (Liu et al., 2008), and Local Outlier Factor (LOF) (Breunig et al., 2000) to determine how easily the generated adversarial upstream forecasts can be detected (Table 3). As generating extreme events such as TCs typically requires larger perturbations, they are easier to be detected. Nevertheless, *Cyc-Attack* is still harder to detect than the baselines. (3) We examine if the adversarial upstream forecasts can be utilized by gradient-based methods (PGD) to generate adversarial input $\mathbf{X}'$. As shown in Figure 6, perturbing upstream *GraphCast* inputs can successfully alter downstream forecasts. Supplementary experimental results are provided in Appendix D.

Table 3: Results of applying 3 anomaly detection methods against adversarial upstream forecasts generated by different attack methods on TC dataset. *Precision* is the fraction of detected anomalies that are truly adversarial; *Recall* is the fraction of adversarial samples correctly detected; and *F1-score* is the harmonic mean of precision and recall. Smaller values imply higher attack effectiveness.

| Defensor | Attackers | Precision↓ | Recall↓ | F1-score↓ |
|---|---|---|---|---|
| PCA-based anomaly detection | Cyc-Attack | 0.9793 | 0.8711 | 0.9220 |
| | ALA | 0.9794 | 1.0000 | 0.9896 |
| | TAAOWPF | 0.9896 | 1.0000 | 0.9948 |
| | AOWF | 0.9922 | 1.0000 | 0.9961 |
| Isolation Forest (IF) | Cyc-Attack | 1.0000 | 0.8447 | 0.9158 |
| | ALA | 1.0000 | 1.0000 | 1.0000 |
| | TAAOWPF | 0.9974 | 1.0000 | 0.9987 |
| | AOWF | 0.9974 | 1.0000 | 0.9987 |
| Local Outlier Factor (LOF) | Cyc-Attack | 0.9970 | 0.8605 | 0.9237 |
| | ALA | 0.9974 | 1.0000 | 0.9987 |
| | TAAOWPF | 0.9974 | 1.0000 | 0.9987 |
| | AOWF | 0.9974 | 1.0000 | 0.9987 |

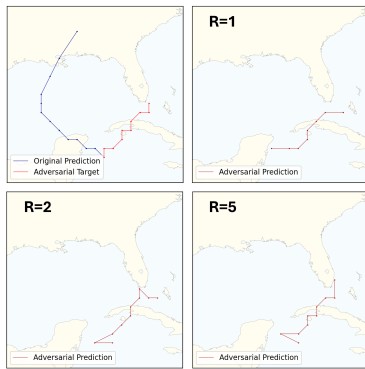

Figure 5: Hurricane *Delta* (from 10/26/2020 to 11/05/2020). Description follows Figure 4, except here we compare the effect of dilating the adversarial target with radii $R = \{1, 2, 5\}$ during attacks.

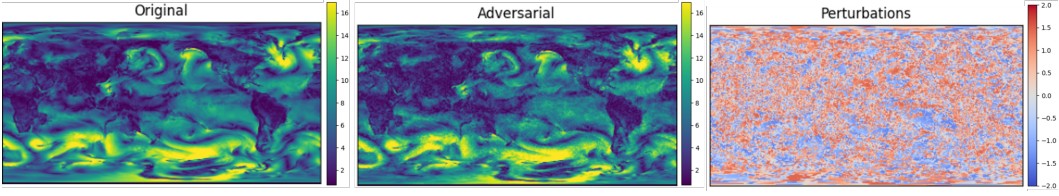

Figure 6: Perturbations (right) were added to the original *GraphCast* wind-speed inputs (left, m/s) 12 hours before the starting date of *Hurricane Delta* on 10/26/2020. The resulting adversarial inputs (middle, generated by the standard PGD (Madry et al., 2017)) led *GraphCast* to generate forecasts that produced the manipulated downstream TC trajectory shown in Figure 5 for $R = 1$.

## 6 DISCUSSION AND ETHICS STATEMENT

This work demonstrates the feasibility of adversarially perturbing upstream DLWF forecasts to steer downstream TC trajectory predictions toward an attacker's chosen target. This result underscores the sensitivity of downstream applications to subtle and imperceptible changes in upstream forecasts. In practice, upstream forecast providers and downstream users are often separate entities, making the integrity and validation of shared data essential. **Our work does not target a specific operational system.** The goal is to highlight potential vulnerabilities and motivate the development of more robust DLWF models as well as security-aware practices across the forecasting pipeline.

## 7 CONCLUSIONS

This paper presents *Cyc-Attack*, a novel adversarial attack method for manipulating weather forecasts generated by DLWF models to alter the trajectory paths produced by TC detection systems. *Cyc-Attack* trains a differentiable surrogate model to enable gradient-based attacks, incorporates a skewness-aware loss with kernel dilation to address class imbalance, and applies distance-based weighting with regularization to ensure perturbations remain realistic and imperceptible. These design choices overcome key limitations of standard adversarial attack approaches under this scenario, allowing *Cyc-Attack* to outperform various baselines with higher targeted TC detection rates and lower false-alarm rates while producing stealthier perturbations. Future work includes developing strategies to robustify the DLWF models against such adversarial attacks.

## REPRODUCIBILITY STATEMENT

For reproducibility, our code is available at the anonymous repository: `https://anonymous.4open.science/r/Cyc-Attack-23855/Submission_Adversarial_Attacks.ipynb`. In Appendix C.1, we provide details on the datasets and their preprocessing. Appendix C.2 presents the implementation details of surrogate model pre-training to ensure reproducibility. Section 5.1 describes the implementation details of the adversarial attack methods.

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

## A   LLM Usage Statement

The use of LLMs in this work was limited solely to minor grammar and wording improvements on the authors' original content.

## B   TempestExtremes Software for TC trajectory prediction

*TempestExtremes* (Ullrich et al., 2021), which is a black-box software built on physical rules for TC trajectory detection, operates in two stages: (1) the first stage is to identify candidate locations as local minima in *mean sea level pressure* (MSL) that are not within $6°$ of a deeper MSL minimum, enclosed by a 200 hPa closed contour within a $5.5°$ radius; and co-located with maxima in the *geopotential thickness* field computed as the difference in geopotential height between 300 and 500 hPa (Z300–Z500), where the maxima are enclosed by a $58.8$ m$^2$s$^{-2}$ closed contour within a $6.5°$ radius, allowing stronger peaks within $1.0°$. The variables associated with each candidate location, including *mean sea level pressure*, *elevation*, and the maximum regional *10-meter wind speed*, are also the output of the first stage. (2) The second stage is to stitch these candidate locations into trajectories constrained by a maximum $8°$ step distance, a maximum 24-hour gap, a minimum 54-hour lifetime, and at least 10 time steps with *10-meter wind speed* no less than 10 m/s, *elevation* not exceeding 150 m, and *latitude* between $-50°$ and $50°$.

In short, the input to *TempestExtremes* includes the weather variables *mean sea level pressure*, *10-meter wind speed*, *elevation*, and *geopotential thickness*, and the final output includes the TC trajectories consisting of stitched candidate locations within a time period.

## C   Details on experimental settings

### C.1   Datasets and Preprocessing

Three pre-trained checkpoints of the *GraphCast* model (Lam et al., 2023) have been released by Google DeepMind. They differ in data sources (*ERA5* or *ERA5+HRES*), spatial resolution ($0.25°$ or $1.0°$), the number of pressure levels (37 or 13), mesh configurations (*2to6* or *2to5*), and whether *precipitation* is used only as an output or as both input and output. Given the model-agnostic nature of adversarial attacks, it is not constrained by the specific architecture of the attacked model. In this study, we used the checkpoint trained on *ERA5*, with $1.0°$ resolution, 13 pressure levels, mesh *2to5*, and precipitation included as both input and output[2].

The input ERA5 data of the *GraphCast* model includes: (1) five single-level variables — *2-meter temperature* (K), *mean sea level pressure* (Pa), *10-meter u-component of wind* (m/s), *10-meter v-component of wind* (m/s), and *total precipitation* (m); (2) six multi-level variables — *temperature* (K), *geopotential* (m$^2$/s$^2$), *u-component of wind* (m/s), *v-component of wind* (m/s), *vertical velocity* (Pa/s), and *specific humidity* (kg/kg), provided at thirteen pressure levels (50, 100, 150, 200, 250, 300, 400, 500, 600, 700, 850, 925, and 1000 hPa); (3) three forcing variables — *total-of-atmosphere incident solar radiation* (J/m$^2$), *land-sea mask* (unitless), and *surface geopotential* (m$^2$/s$^2$); and (4) four post-calculated variables — *sine and cosine of year progress*, and *sine and cosine of day progress*, which are derived from timestamps and are unitless.

The output data of the *GraphCast* model includes the five single-level variables and six multi-level variables described above. All data are provided at a 6-hour temporal resolution, corresponding to 00Z, 06Z, 12Z, and 18Z each day, with 2 time steps used as input to the *GraphCast* model, and 1, 4, 12, 20, or 50 time steps used as forecast outputs. The input and output data are not standardized, as there is a built-in standardization process within the *GraphCast* model. Besides, the hourly precipitation values in ERA5 data are aggregated over six-hour windows.

The input to the *TempestExtremes* (Ullrich et al., 2021) includes the weather variables *mean sea level pressure*, *10-meter wind speed*, *elevation*, and *geopotential thickness*. Among these variables, *wind speed* and *geopotential thickness* are not directly available in the *GraphCast* forecast output, but can be derived. *Wind speed* is computed using the square root of the sum of squares of the u-

---

[2]The selected checkpoint is *"GraphCast_small - ERA5 1979-2015 - resolution 1.0 - pressure levels 13 - mesh 2to5 - precipitation input and output.npz"*.

and v-component winds, while *geopotential thickness* is obtained by taking the difference between the geopotential heights at 300 hPa and 500 hPa (i.e., Z300-Z500). Additionally, *elevation* can be approximated by dividing the surface geopotential by 9.08665, where the surface geopotential is obtained from external datasets.

## C.2 Implementation Details of Pretraining the surroate model

Serving as a proxy for the black-box *TempestExtremes* software, the surrogate models' input data are aligned with that of *TempestExtremes*. This input consists of weather variables—*mean sea level pressure*, *geopotential thickness*, *10-meter wind speed*, and *elevation*—extracted from the constructed ground truth ERA5 dataset across 12 time steps within the forecast window of the *Graph-Cast* model. All training inputs are standardized using the variable-wise mean and standard deviation computed from our collected dataset. The output corresponds to a dilated region centered at the trajectory locations detected by the black-box *TempestExtremes* software at each of the 12 time steps, covering a $r$-hop neighborhood around each trajectory location. For training, we randomly partition the input–output pairs into training, validation, and test sets using a 0.6:0.2:0.2 split, resulting in 2052 training samples, 684 validation samples, and 684 test samples after discarding any pairs containing undetected nodes.

We use the *Adam* optimizer (Kingma, 2014), and perform hyperparameter tuning using *Optuna* (Akiba et al., 2019), an automatic optimization framework. A total of 30 trials are conducted to search for optimal values of the following hyperparameters: *learning rate* in the range $[1e-5, 1e-2]$, $\beta_1$ in $[0.5, 0.99]$, $\beta_2$ in $[0.9, 0.9999]$, *weight decay* in $[1e^{-6}, 1e^{-3}]$, *convolution stride size* from the set $\{1, 2, 3, 4, 6, 8\}$, and *number of epochs* in the range $[10, 100]$ for training the *DeepLabV3+* model. For each trial, the model achieving the best validation performance is retained. The optimal hyperparameters—*learning rate* $= 5.92 \times 10^{-4}$, $\beta_1 = 0.9101$, $\beta_2 = 0.9119$, weight decay $= 6.48 \times 10^{-6}$, *stride size* = 1, and *number of epochs* = 77—were identified in trial 1, yielding a best validation loss of 16.72.

The pre-trained surrogate model produces outputs by taking the same inputs as the *TempestExtremes* software. However, unlike *TempestExtremes*, the surrogate model does not output additional variables associated with each detected location, such as *mean sea level pressure*, *elevation*, and the regional maximum *10-meter wind speed*. For each location detected by the surrogate models, the variables *mean sea level pressure* and *elevation* are directly taken from the corresponding weather data at that node. The maximum regional *10-meter wind speed*, in contrast, is constructed manually. Let $\lambda_{i_0 j_0}$ and $\phi_{i_0 j_0}$ denote the *longitude* and *latitude*, respectively, of a candidate location $(i_0, j_0)$, in degrees; and let $w_{i_0 j_0}$ denote the *10-meter wind speed* at $(i_0, j_0)$. A candidate location is considered missing if $w_{i_0 j_0} = \text{NaN}$ or if $|w_{i_0 j_0} - 10^{20}| < 10^{-6}$. The maximum regional *10-meter wind speed* is constructed as follows. (i) For each candidate location $(i_0, j_0)$, map its coordinates $(\lambda_{i_0 j_0}, \phi_{i_0 j_0})$ to $(\Lambda_{i_0 j_0}, \Phi_{i_0 j_0})$ in radians as $\Lambda_{i_0 j_0} = \left( \frac{\lambda_{i_0 j_0}}{180} \pi \right) \bmod (2\pi)$, $\Phi_{i_0 j_0} = \frac{\phi_{i_0 j_0}}{180} \pi$, where $\Lambda_{i_0 j_0} \in [0, 2\pi)$. (ii) For each candidate location $(i_0, j_0)$, compute the central angle $\Delta \sigma_{ij}$ in radians between $(i_0, j_0)$ and any point $(i, j)$ as $\Delta \sigma_{ij} = \arccos(c_{ij})$, where $c_{ij} = \sin \Phi_{i_0 j_0} \sin \Phi_{ij} + \cos \Phi_{i_0 j_0} \cos \Phi_{ij} \cos(\Lambda_{ij} - \Lambda_{i_0 j_0})$, with $c_{ij}$ clipped to the interval $[-1, 1]$ such that $\Delta \sigma_{ij}$ is clipped to the interval $[0, \pi]$. The corresponding great-circle distance between $(i, j)$ and $(i_0, j_0)$ is then computed as $D_{ij} = \frac{180}{\pi} \Delta \sigma_{ij}$. (iii) For the candidate location $(i_0, j_0)$, define the set of neighboring nodes located along eight directions (N, NE, E, SE, S, SW, W, NW) and within a great-circle distance of 2 as $\mathcal{S}_{(i_0, j_0)} = \{(i, j) \mid D_{ij} \leq 2; w_{ij} \text{ is not missing}\}$. (iv) If $\mathcal{S}_{(i_0, j_0)}$ is nonempty, define the maximum regional *10-meter wind speed* at candidate location $(i_0, j_0)$ as $\max_{(i,j) \in \mathcal{S}_{(i_0, j_0)}} w_{ij}$; otherwise, it is defined as $w_{i_0 j_0}$.

Notice that due to subtle differences in floating-point precision between Python (used in the reconstruction process above) and C++ (used in *TempestExtremes*), discrepancies may occur near the threshold distance $D_{ij} = 2.0$. Specifically, some neighbors identified by *TempestExtremes* as within $D_{ij} = 2.0 - \epsilon$ may be computed as $D_{ij} = 2.0 + \epsilon$ in Python, and thus excluded, and vice versa, where $\epsilon$ is a very small error. To resolve this, we uniformly set the threshold to $D_{ij} = 2.0 + \epsilon$ with $\epsilon = 10^{-8}$ in both implementations. This ensures that the computed maximum regional *10-meter wind speed* exactly matches each from *TempestExtremes*.

## C.3 IMPLEMENTATION DETAILS OF CONSTRUCTING ADVERSARIAL TARGETS

We propose a strategy for constructing adversarial downstream targets. Given the original downstream prediction $\hat{\mathbf{Z}} \in \{0,1\}^{\beta \times r \times c}$, we treat each contained trajectory as a sequential path $\hat{\mathbf{p}} = \{\hat{\mathbf{p}}_\tau \in \mathbb{R}^2 \mid \tau = 1, \ldots, \beta\}$, where $\hat{\mathbf{p}}_\tau = (\lambda_\tau, \phi_\tau)$ denotes the longitude and latitude (in degrees) at time step $\tau$. Our goal is to construct an adversarial trajectory $\hat{\mathbf{p}}' = \{\hat{\mathbf{p}}'_\tau \in \mathbb{R}^2 \mid \tau = 1, \ldots, \beta\}$ that preserves the temporal step length of the corresponding original one while progressively deviating in direction. Then, the constructed adversarial trajectory will be converted into adversarial downstream targets $\hat{\mathbf{Z}}^* \in \{0,1\}^{\beta \times r \times c}$.

In detail, the adversarial trajectory $\hat{\mathbf{p}}'$ begins at the same origin as the original trajectory $\hat{\mathbf{p}}$, i.e., $\hat{\mathbf{p}}'_1 = \hat{\mathbf{p}}_1$. For each subsequent step $\tau > 1$, we compute the great-circle distance $\hat{d}_\tau$ between $\hat{\mathbf{p}}_{\tau-1}$ and $\hat{\mathbf{p}}_\tau$ using the spherical law of cosines by

$$\cos \Delta\sigma_\tau = \sin \Phi_{\tau-1} \sin \Phi_\tau + \cos \Phi_{\tau-1} \cos \Phi_\tau \cos(\Lambda_\tau - \Lambda_{\tau-1}),$$

$$\hat{d}_\tau = R \cdot \arccos\big(\min(\max(\cos \Delta\sigma_\tau, -1), 1)\big),$$

where $(\Lambda_\tau, \Phi_\tau)$ and $(\Lambda_{\tau-1}, \Phi_{\tau-1})$ are the radian representations of the coordinates, and $R$ is the Earth's radius in kilometers.

Let $\mathcal{M} = \{\boldsymbol{\delta}_j \in \mathbb{R}^2 \mid j = 1, \ldots, 8\}$ denote the set of 8-connected compass direction unit vectors on the 2D plane. Each direction $\boldsymbol{\delta}_j$ corresponds to a compass bearing $\theta_j \in [0°, 360°)$. At each time step, we sample a direction index $j^*$ according to a probability distribution derived from angular deviation from the original path.

To quantify directional deviation, we first define the cosine similarity between the original displacement vector and each candidate direction. Let $\hat{\mathbf{v}}_\tau = \hat{\mathbf{p}}_\tau - \hat{\mathbf{p}}_{\tau-1}$ be the original vector, and let $\hat{\mathbf{v}}'_{\tau-1} = \hat{\mathbf{p}}'_{\tau-1} - \hat{\mathbf{p}}'_{\tau-2}$ be the previous adversarial displacement when $\tau > 2$. We compute

$$\cos \theta_j^{\text{orig}} = \frac{\hat{\mathbf{v}}_\tau^\top \boldsymbol{\delta}_j}{\|\hat{\mathbf{v}}_\tau\|}, \quad \cos \theta_j^{\text{adv}} = \frac{\hat{\mathbf{v}}'^\top_{\tau-1} \boldsymbol{\delta}_j}{\|\hat{\mathbf{v}}'_{\tau-1}\|},$$

which measures the deviation magnitude from the current original step and the coherence with the prior adversarial step, respectively. Then, we assign a score $s_j$ to each direction $\delta_j$ by

$$s_j = \gamma_1 \cdot \exp(-\cos \theta_j^{\text{orig}}) + \gamma_2 \cdot \exp(\cos \theta_j^{\text{adv}}).$$

These scores are normalized into probabilities $P_j = s_j / \sum_{\ell=1}^8 s_\ell$, and the direction $j^*$ with the highest probability is selected.

To ensure smooth and plausible deviation, we compute a blended unit direction vector by summing the original direction $\hat{\mathbf{v}}_\tau$ and the selected direction $\boldsymbol{\delta}_{j^*}$ by

$$\mathbf{u}_\tau = \frac{\hat{\mathbf{v}}_\tau + \boldsymbol{\delta}_{j^*}}{\|\hat{\mathbf{v}}_\tau + \boldsymbol{\delta}_{j^*}\|},$$

and then convert it to a compass bearing $\theta_\tau$ by

$$\theta_\tau = \text{atan2}(u_{\tau,0}, u_{\tau,1}),$$

converted to degrees in $[0°, 360°)$.

The next adversarial trajectory location is obtained by projecting from the previous position $\hat{\mathbf{p}}'_{\tau-1} = (\lambda'_{\tau-1}, \phi'_{\tau-1})$ a geodesic distance $\hat{d}_\tau$ along bearing $\theta_\tau$, using the following great-circle formulas

$$\phi'_\tau = \arcsin\left(\sin \phi'_{\tau-1} \cos \frac{\hat{d}_\tau}{R} + \cos \phi'_{\tau-1} \sin \frac{\hat{d}_\tau}{R} \cos \theta_\tau\right),$$

$$\lambda'_\tau = \lambda'_{\tau-1} + \arctan 2\left(\sin \theta_\tau \sin \frac{\hat{d}_\tau}{R} \cos \phi'_{\tau-1}, \cos \frac{\hat{d}_\tau}{R} - \sin \phi'_{\tau-1} \sin \phi'_\tau\right),$$

where all angles are in radians and $R$ is the Earth's radius in kilometers. By iterating this process from $\tau = 2$ to $\beta$, we obtain an adversarial trajectory $\hat{\mathbf{p}}' = \{\hat{\mathbf{p}}'_\tau \in \mathbb{R}^2 \mid \tau = 1, \ldots, \beta\}$ that

preserves geodesic step lengths while progressively deviating in direction from the original path $\hat{\mathbf{p}} = \{\hat{\mathbf{p}}_\tau \in \mathbb{R}^2 \mid \tau = 1, \dots, \beta\}$.

For each original downstream prediction $\hat{\mathbf{Z}}$, if it contains at least one detected trajectory, it will be included in our pool of valid samples for constructing adversarial targets. Otherwise, it will be discarded. As a result, among all $285$ samples, we retain $149$ valid samples. For each valid sample, if it contains $N$ detected trajectories, we construct $N$ adversarial samples by targeting one trajectory at a time, while keeping the other $N-1$ trajectories unchanged. This yields $N$ adversarial versions per valid sample, with one adversarial trajectory constructed in each.

Finally, we construct each adversarial downstream $\hat{\mathbf{Z}}^* \in \mathbb{R}^{\beta \times r \times c}$ from the original one $\hat{\mathbf{Z}} \in \mathbb{R}^{\beta \times r \times c}$ by replacing the trajectory locations along the original predicted trajectory with those along the constructed adversarial trajectory, while leaving all others unchanged.

## D  SUPPLEMENTARY EXPERIMENTAL RESULTS

Figure 7 provides supplementary results illustrating the sensitivity of *Cyc-Attack* to the choice of dilation radius. Figure 8 offers additional visualizations for comparing the performance of different adversarial attack methods.

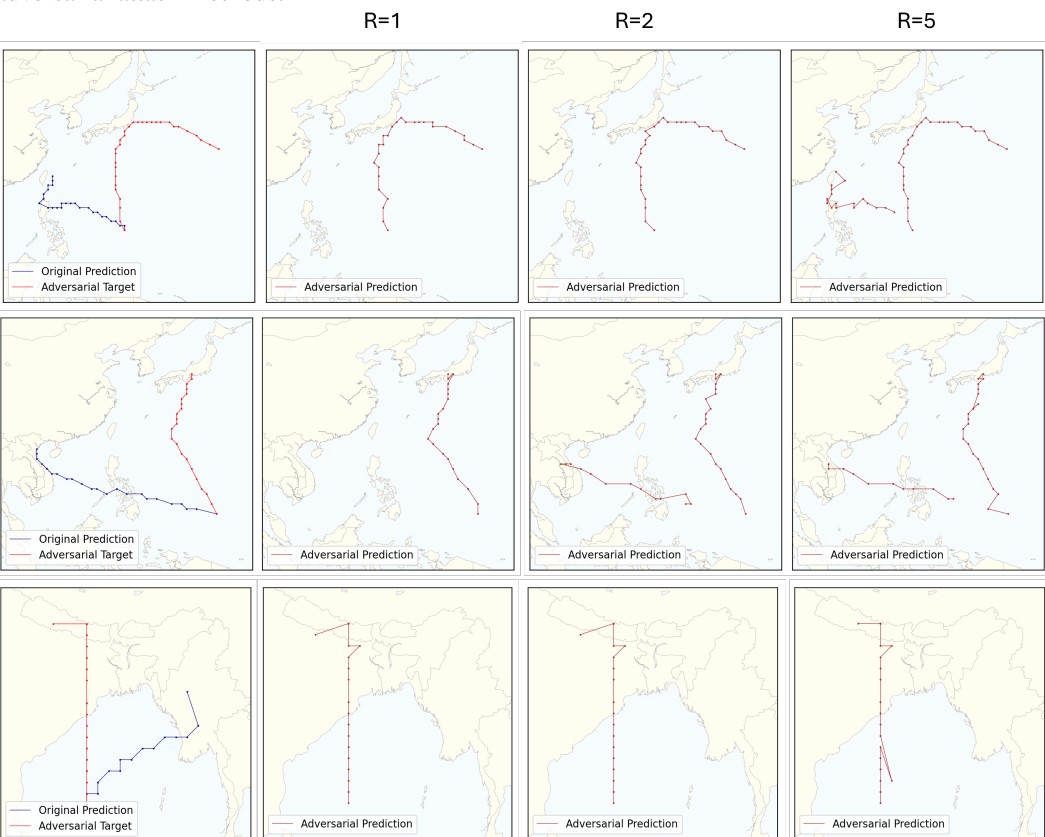

Figure 7: Cyclone *Maria* (09/16/2017–09/26/2017), Cyclone *Haiyan* (11/03/2013–11/13/2013), and Cyclone *Nargis* (04/27/2008–05/07/2008), shown from top to bottom. The description follows Figure 5.

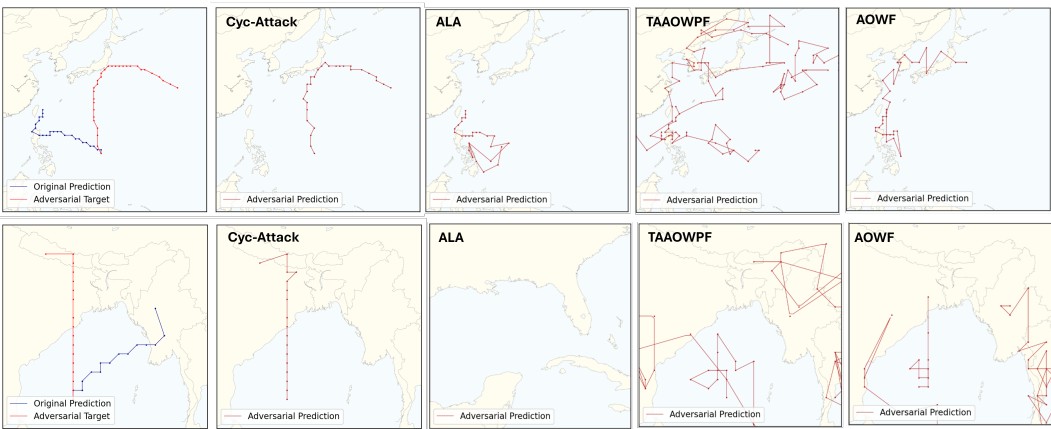

Figure 8: Cyclone *Maria* (09/16/2017–09/26/2017) and Cyclone *Nargis* (04/27/2008–05/07/2008), shown from top to bottom. The description follows Figure 4.

