# OpenReview forum: "Adversarial Attacks on Downstream Weather Forecasting Models: Application to Tropical Cyclone Trajectory Prediction"
_ICLR.cc/2026/Conference — Submitted to ICLR 2026_

### Official Review · Reviewer_cu9v · 2025-10-26

**Soundness:** 2
**Presentation:** 3
**Contribution:** 2
**Rating:** 4
**Confidence:** 3

**Summary:**

This paper introduces Cyc-Attack, an adversarial attack method that manipulates deep learning-based weather forecasts to alter downstream tropical cyclone trajectory predictions. The paper focuses on addressing the black-box nature of cyclone detectors and the extreme class imbalance issue by employing a differentiable surrogate model, a skewness-aware loss with kernel dilation, and distance-based gradient weighting to generate stealthy and effective perturbations.

**Strengths:**

1. It is the first work to demonstrate how to attack a downstream weather application (cyclone tracking) by perturbing upstream forecasts.

2.  The method outperforms several baselines in terms of both attack success rate (higher trajectory detection rate) and stealth (lower false alarm rate and smaller perturbations).

**Weaknesses:**

1.  The attack's effectiveness may highly rely on the accuracy of the pre-trained surrogate model in approximating the black-box detector; inaccuracies here could degrade performance.

2. The process involves pre-training a surrogate model and running an iterative, gradient-based attack, which can be computationally expensive

3. The technical contribution of the paper is a little limited as adversarial attack on time series problems has been well studied.

4. The real-world impact of the proposed method is not very clear since how an attacker can add the perturbation in practice is not very clear.

**Questions:**

Please refer to the weakness part.

---

> ### Author Response · Authors · 2025-12-04
>
> **1. Weakness 1.** We thank the reviewer for raising this point, as it leads to a more in-depth discussion. We acknowledge that the attack’s effectiveness may rely on the accuracy of the pre-trained surrogate model in approximating the black-box detector, and that inaccuracies could degrade performance. However, **using a surrogate model is one of the most common and standard approaches** for adversarial attacks in black-box settings, where direct access to the target model is unavailable. As such, this reliance on surrogate approximation is **a general challenge of black-box attacking settings and is not unique to our method**. **To address this challenge**, we incorporate domain-specific designs (e.g., skewness-aware loss and kernel dilation) to improve the surrogate’s reliability under severe class imbalance for our problem. Our experiments show that **the proposed attack remains effective even when the surrogate does not perfectly match the black-box detector**, indicating robustness to reasonable surrogate inaccuracies. Specifically, as shown by the accuracy of the surrogate model in Table 1, when R = 2, the True Positive Rate for TC node classification reaches 0.8131 (with 1.0 being optimal), while the False Negative Rate is 0.1869 (with 0.0 being optimal). Despite these imperfect classification results, the experimental outcomes in Table 2 and Figure 4 consistently demonstrate the effectiveness of our proposed method.
>
> **2. Weakness 2.** We thank the reviewer for giving us the opportunity to clarify the computational expense associated with pre-training a surrogate model and running an iterative, gradient-based attack. First, **the use of iterative gradient-based optimization is one of the most commonly adopted strategies** in adversarial attacks and is widely considered a standard practice in this field. In general, the number of optimization steps can be explicitly limited as a trade-off, **which helps reduce the overall computational cost**. Second, in black-box settings where direct gradient access is unavailable, **training a surrogate model is a necessary step** to enable the implementation of gradient-based adversarial attacks. It is worth noting that surrogate model **pre-training is performed offline and only once**, after which the trained model can be reused for multiple attack instances.
>
> **3. Weakness 3.** We thank the reviewer for the feedback. However, **we respectfully disagree** that the technical contribution of the paper is a little limited, as adversarial attacks on time series problems have been well studied, for the following reasons. (a) **Beyond time series problems**, this paper contributes to adversarial attacks for geospatial-temporal problems, which require not only temporal realisticness (as in time series problems) but also spatial realisticness across geographic fields. The joint consideration of spatial and temporal realisticness in adversarial attacks for weather forecasting systems **has not been well studied**. (b) For adversarial attacks in geospatial-temporal weather forecasting systems, one of the key challenges identified in this paper is **ensuring spatial realisticness**, such that the adversarially generated forecasts remain close to the original outputs while producing realistic TC trajectories that are free of conspicuous artifacts (e.g., zigzag paths) that would make the attack easily detectable. To address this challenge, this paper contributes technical solutions and demonstrates improved performance over existing methods.
>
> **4. Weakness 4.** We thank the reviewer for pointing out that a clearer description of how an attacker can add the perturbation in practice is expected. **In the section on Discussion and Ethics Statement, we briefly noted** that “in practice, upstream forecast providers and downstream users are often separate entities, making the integrity and validation of shared data essential.” **We now expand this to make it clearer as follows**. (a) Modern weather forecasting systems rely on observational data collected from a highly decentralized and heterogeneous network of sources, which operate under different jurisdictions and institutional incentives. **This fragmented data ecosystem creates a broad attack surface, providing adversaries with multiple potential entry points to tamper with observations before they are assimilated into forecasting pipelines [1]**. (b) Data assimilation explicitly models uncertainty arising from both observation error and background error. Since the assimilated state is designed to tolerate noise within statistically expected variance ranges, carefully crafted manipulations can be concealed within this uncertainty envelope. As noted by [1], **this inherent uncertainty “plays into the attacker’s hand,”**.
>
> [1] Imgrund, E., Eisenhofer, T., \& Rieck, K. (2025). Adversarial Observations in Weather Forecasting. arXiv preprint arXiv:2504.15942.

---

### Official Review · Reviewer_QRZv · 2025-10-26

**Soundness:** 3
**Presentation:** 2
**Contribution:** 3
**Rating:** 6
**Confidence:** 3

**Summary:**

This paper proposes Cyc-Attack, a novel adversarial attack method that perturbs the outputs of upstream weather forecasting models to manipulate downstream tropical cyclone trajectory predictions. The method effectively addresses challenges such as data sparsity and extreme class imbalance through the use of a differentiable surrogate model, skewness-aware loss, kernel dilation, and distance-based gradient weighting, enabling successful attacks in black-box settings while generating realistic and stealthy adversarial trajectories.

**Strengths:**

1、The focus on tropical cyclone trajectory prediction provides a highly relevant and impactful setting for studying the vulnerabilities of deep learning weather forecasting models in downstream applications, offering valuable insights for improving real-world robustness.

2、The paper systematically addresses several critical challenges, including the non-differentiability of black-box TC detectors, severe class imbalance, premature attack termination due to surrogate model errors, and the generation of unrealistic zigzag trajectories, demonstrating technical depth and completeness.

**Weaknesses:**

1、Several critical choices, especially in experimental design and parameter selection, lack in-depth justification, which affects the clarity and persuasiveness of the paper. Specific issues are listed below under "Questions."

**Questions:**

1、In Section 5.2, when R=0, FPR is 0.0067 and TPR is 0.9896; when R=2, FPR drops to 0.0002, but TPR decreases significantly to 0.8131. Why is the reduction in FPR from 0.0067 to 0.0002 considered more important than the drop in TPR from 0.9896 to 0.8131? Has the risk trade-off between false alarms and missed detections been evaluated in the context of meteorological early warning systems? Is this choice supported by domain expertise or operational requirements?
2、In the first row of Figure 4, the adversarial trajectories generated by baseline methods (e.g., ALA, TAAOWPF, AOWF) are not shown. Is this because these methods failed to produce complete and coherent trajectories, making visualization impossible? If so, does this indicate that these methods are ineffective at the trajectory level? The authors should clarify this in the text and, if possible, include their outputs in the appendix for a more comprehensive comparison.

---

> ### Author Response · Authors · 2025-12-04
>
> **1. Question 1.** We thank the reviewer for this insightful question. From $R=0$ to $R=2$, FPR is reduced by about 97.0\% (from 0.0067 to 0.0002), while TPR decreases by about 17.8\% (from 0.9896 to 0.8131), reflecting that the reduction in FPR is considered more important than the drop in TPR.
>
> **This trade-off can be interpreted by both domain expertise and operational requirements.** In meteorological early warning systems, TC detection suffers from a **severe imbalance issue**, where TC-affected nodes are extremely rare. In such settings, **even a small FPR can lead to a large number of false alarms**, which may cause unnecessary responses, alarm fatigue, and reduced public trust. Figure 7 in Appendix D further illustrates that higher FPRs are associated with scattered or spurious TC detections, which are particularly undesirable in operational practice. Therefore, **to prioritize the reduction of false alarms is critical for maintaining system reliability**. In our paper, $R=2$ is chosen because it represents a better practical trade-off as evidenced in Table 1: compared with $R=0$, it substantially reduces FPR while **still maintaining a reasonably high TPR (0.8131)**, and compared with larger $R$, it avoids excessive degradation in detection performance and trajectory realism. **This can be further supported by the NHC 2025 Verification Report (Figure 5) [1].** The official forecast achieves a trajectory-level probability of detection around 0.6 while maintaining a substantial **1 − FAR ≈ 0.85**, depending on storm type and lead time. These reported values highlight that even state-of-the-art operational systems accept non-negligible missed-event rates in exchange for suppressing false alarms, due to the high societal and economic costs associated with unnecessary warnings. These operational statistics reinforce that our chosen operating point (lower FPR while retaining TPR > 0.8) is well-aligned with professional forecasting priorities and realistic performance levels observed in practice.
>
> [1] Cangialosi, J., & Martinez, J. (2025). 2024 NHC Verification Report Preview: Atlantic Basin.
>
> **2. Question 2.** We thank the reviewer for giving us the opportunity to clarify this point. In the first row of Figure 4, the adversarial trajectories generated by the baseline methods (e.g., ALA, TAAOWPF, and AOWF) are not shown because (a) these methods are capable of **effectively removing the original TC trajectories (blue)** from the downstream detection results, yet (b) they **fail to induce the downstream physical rule-based software to produce complete and coherent adversarial TC trajectories (red)**, making visualization impossible. **Yes, our observations indicate that these baseline methods can be ineffective at the trajectory level in some cases.** For example, as shown in Fig. 4, under the Hurricane Delta scenario, the baseline methods are able to remove the original TC trajectory but fail to produce the targeted adversarial trajectory. Under the Typhoon Haiyan scenario, some baselines are unable to fully remove the original trajectory and, in addition, produce additional undesired trajectories. We also include **more outputs of these baseline methods in Appendix D** for a more comprehensive qualitative comparison.

---

### Official Review · Reviewer_FURh · 2025-10-31

**Soundness:** 3
**Presentation:** 4
**Contribution:** 3
**Rating:** 8
**Confidence:** 3

**Summary:**

This paper studies the vulnerability of deep learning–based weather forecasting systems to adversarial attacks, focusing on the downstream task of tropical cyclone trajectory prediction. Overall it is a timely and well-executed study on adversarial vulnerabilities in downstream weather forecasting pipelines. The method is well-motivated and empirically validated. While the evaluation scope and realism analysis could be expanded, the work is technically sound, clearly written, and potentially impactful for both the adversarial ML and climate science communities.

**Strengths:**

1. Overall the paper is well-written and easy to follow. The algorithm design is also technically sound.
2. The paper extends adversarial robustness analysis from general DLWF models to downstream applications like TC trajectory prediction, which is societally relevant and technically distinct from prior pixel-level attacks.
3. The study uses real-world datasets (ERA5, IBTrACS) and provides thorough quantitative comparisons, ablations, and visualization (e.g., Hurricane Delta, Typhoon Haiyan). Metrics at both location and trajectory levels are carefully defined.

**Weaknesses:**

1. While the authors constrain perturbations via distance weighting, physical realism is not formally validated. The “stealthiness” metric is purely statistical (ℓ1 distance).
2. It seems that the defenses discussed in the paper are more of detections rather than technques that can make the forecasting models more robust to attacks. Would appreciate more discussions on the defense side.

**Questions:**

Please see weaknesses.

---

> ### Author Response · Authors · 2025-12-04
>
> **1. Weakness 1.** We thank the reviewer for this valuable comment. We agree that the current stealthiness metric based on the $\ell_1$ distance primarily reflects statistical similarity. A more comprehensive assessment of physical realism would ideally verify whether the perturbed forecast still adheres to key atmospheric principles, such as continuity and coherent motion in the trajectory.
>
> Although **formal physics-realism evaluation remains an open problem** for high-dimensional DL-based weather forecasts, **our method includes mechanisms that promote physical plausibility**. In particular, the distance-based gradient weighting and regularization constrain perturbations and suppress irrelevant distortions, helping maintain realistic TC trajectories free of conspicuous artifacts such as zigzag paths. These designs ensure that the manipulated trajectories retain smooth and plausible motion consistent with real TC behavior, **which can be evaluated by visualizations,** as shown in Figures 4, 7, and 8 in our paper.
>
> We appreciate the reviewer’s suggestion and **consider the development of more rigorous physics-aware stealthiness measures a promising future direction.**
>
> **2. Weakness 2.** We thank the reviewer for this valuable comment. We agree that the defenses discussed in the paper are more of detections rather than techniques that can make the forecasting models more robust to attacks. This is because **the primary goal of this work is to highlight potential vulnerabilities and demonstrate the feasibility** of adversarial manipulation in downstream weather forecasting tasks.
>
> **In the section on Conclusion, we noted** that future work includes developing strategies to robustify the DLWF models against such adversarial attacks. Here, **we would like to provide more discussion on the defense side.** Specifically, our proposed method can be used to construct adversarial samples that are capable of significantly altering system outputs through small perturbations. These adversarial samples can serve as part of the training data to support **adversarial learning/training**, thereby helping make the forecasting models more robust to attacks during the training process. This perspective is consistent with recent surveys [1] in adversarial machine learning that categorize defense strategies into adversarial training and other approaches.
>
> [1] Abomakhelb, A., Jalil, K. A., Buja, A. G., Alhammadi, A., & Alenezi, A. M. (2025). A Comprehensive Review of Adversarial Attacks and Defense Strategies in Deep Neural Networks. Technologies, 13(5), 202.

---

### Official Review · Reviewer_XQBN · 2025-11-01

**Soundness:** 1
**Presentation:** 3
**Contribution:** 1
**Rating:** 2
**Confidence:** 5

**Summary:**

This submission focuses on adversarial attacks against weather forecasting models. It identifies three key research challenges: the black-box constraints of tropical cyclone (TC) detection systems, the sparsity of TC events, and the need to maintain physical consistency in the generated perturbations. To address these challenges, the paper proposes Cyc-Attack, a surrogate model–based black-box adversarial attack framework designed to effectively evaluate and exploit the vulnerabilities of TC forecasting systems.

**Strengths:**

**i.** The research topic is interesting and important. While adversarial attacks have been extensively studied in static domains such as image and text classification, their impact on dynamic applications like time series forecasting remains underexplored.

**ii.** The paper is easy-to-follow.

**iii.** The setting that integrates time series forecasting with a downstream tropical cyclone (TC) detection system is interesting and well-motivated, as it reflects a more realistic and application-driven scenario.

**Weaknesses:**

**i.** The research gaps are not accurately identified. Specifically, the third and fourth “challenges” mentioned in the paper should not be considered unsolved research problems. For example, ensuring that adversarial perturbations are imperceptible and realistic is a fundamental property of adversarial examples, not a unique challenge. Similarly, developing a precise surrogate model is an inherent prerequisite for any surrogate model–based black-box attack, rather than a novel research gap. The paper should better clarify which aspects of these challenges are truly new or unexplored in the context of adversarial studies on weather forecasting models.

**ii.** The motivation for adopting a surrogate model–based transfer attack is not sufficiently supported. Prior studies, such as [1] (multi-query) and [2] (one-query), have already proposed zero-order optimization–based black-box attacks for time series forecasting models. Although the manuscript claims that such methods require queries to the black-box system, it does not clearly explain why constructing a surrogate model is preferable to directly querying the true system. This justification is particularly important because the sparsity of tropical cyclone events can significantly degrade the similarity between the surrogate and the target systems, potentially limiting the transferability and practical effectiveness of the proposed approach.

**iii.** The attack pipeline is conceptually unclear. More importantly, the pipeline shown in Figure 2 does not align with the formulation described in Section 4. In this setup, the weather forecast serves as an intermediate output between the deep-learning-based forecasting model and the downstream tropical cyclone (TC) detection system. However, the paper defines the attack as manipulating the prediction \(Y\) rather than the input \(X\), which is inconsistent. If the attack manipulates the forecast output \(Y\) as described in Section 4, then the forecasting model becomes unnecessary, since one could directly perturb \(Y\) before passing it to the TC detector. Conversely, if the attack perturbs the input \(X\) as shown in Figure 2, then the optimization should compute and update \(X'\) instead of \(Y'\) in Section 4. This inconsistency makes the attack pipeline difficult to interpret and weakens the overall clarity of the experimental design.

**iv.** The technical contribution is relatively marginal. The submission trains a surrogate model to perform a transfer-based black-box attack, with the primary challenge attributed to data imbalance. However, neither the use of surrogate model–based attacks nor the approach to address imbalance represents a novel contribution. Both have been well-studied in prior literature, and the manuscript does not provide sufficient methodological innovation or theoretical advancement beyond existing work.

**v.** The experimental evaluation involves only one tropical cyclone (TC) detection system, which limits the assessment of the proposed method’s generalization capability. Without testing on multiple detection systems or model architectures, it remains unclear whether the surrogate model trained in this study can generalize effectively to other TC detection frameworks or forecasting setups.












**References**

[1] Zhu, Lyuyi, et al. "Adversarial diffusion attacks on graph-based traffic prediction models." IEEE Internet of Things Journal 11.1 (2023): 1481-1495.

[2] Liu, Fuqiang, et al. "Adversarial Vulnerabilities in Large Language Models for Time Series Forecasting." International Conference on Artificial Intelligence and Statistics. PMLR, 2025.

**Questions:**

The most interesting aspect of this submission lies in its attempt to integrate time series forecasting models with downstream decision-making systems within an adversarial framework. However, the current implementation appears to manipulate intermediate prediction values rather than the raw inputs to the forecasting models, which weakens the conceptual alignment between the attack formulation and the intended end-to-end adversarial setting.

**Which to attack (three options)**

I can identify how to compute Y' as in Equation 5, but I cannot find how to compute X', even though it is mentioned in Figure 2. You must choose and state one threat model clearly in the paper:

- **(1) Raw-input attack (recommended):** attacker perturbs the raw observations \(X\) that are fed into the forecasting model. This is the most realistic end-to-end setting (sensor spoofing, data-assimilation tampering).
- **(2) Output/forecast attack:** attacker perturbs the published forecast \(Y\) directly (e.g., intercept/modify forecast products). This is a valid but different threat model and must be justified operationally.
- **(3) Joint attack:** attacker can perturb both \(X\) and \(Y\). If used, consistency between \(X'\) and \(Y'\) must be enforced.

---

> ### Author Response · Authors · 2025-12-04
>
> **1. Weakness iii & Question.** We thank the reviewer for letting us clarify the **misunderstanding** and would like to **kindly remind the reviewer** that **the attack pipeline in Figure 2 is consistent with the formulation in Section 4**. This **is explicitly described** in the high-level pipeline description in Section 4 (Page 4, Lines 191–193), where we state that the adversarial upstream forecast $\hat{Y}'$ can then be fed into existing DLWF attack methods to learn the adversarial input $\hat{X}'$ that yields the manipulated forecast (i.e., the targeted trajectory $\hat{Z}^*$). **Moreover, as clarified** in the discussion of the full pipeline (Page 8, Lines 429–431), the adversarial upstream forecasts produced in the second part of the pipeline can be used to construct the adversarial upstream inputs in the first part. **Figure 6 further illustrates** how $\hat{X}$ can be perturbed in practice once $\hat{Y}'$ is specified.
>
> **2. Weakness i.** We thank the reviewer for bringing attention to this point and would like to **kindly remind the reviewer** that the research gaps and related challenges **are explicitly stated in the manuscript.** As described in the Introduction (Page 1, Lines 054–073), adversarial attacks on downstream forecasting tasks remain largely unexplored. We further identify the open challenges that prior work has not addressed, including the black-box nature of TC detection systems, the extreme class imbalance, and the necessity of maintaining realistic and smooth trajectories without producing zigzag artifacts (Page 1, Lines 074–107). Furthermore, **the manuscript explains** that preserving physical plausibility under adversarial perturbations remains an open problem in DL-based weather forecasting (Page 1, Lines 108–116). Cyc-Attack directly addresses these issues through surrogate modeling, skewness-aware learning, and distance-based perturbation control, which correspond to the contributions summarized in the paper.
>
> **3. Weakeness ii.** We thank the reviewer for giving us the opportunity to clarify and more sufficiently support the motivation for adopting a surrogate model-based transfer attack.
>
> First, we would like to kindly remind the reviewer that the cited ZO-based attacks are designed for traffic or generic multivariate time-series data and **cannot be directly applied to our high-dimensional geospatio-temporal meteorological fields**. The strong spatial–temporal coupling of atmospheric forecasts makes ZO gradient estimation highly inaccurate and extremely query-intensive, resulting in prohibitive computational cost.
>
> Furthermore, TC grids are extremely sparse, so most ZO perturbations do not influence the detection outcome, producing weak and noisy signals that further degrade gradient estimation. **Existing ZO attacks do not account for this level of sparsity.**
>
> By contrast, the surrogate model provides structured gradients focusing on TC-sensitive regions. Empirically, when we adopt a ZO-style update, **each iteration** in the adversarial attacking process requires two black-box calls and is **on average 15 seconds slower per epoch** on a single A100 than our surrogate-based optimization, and more fine-grained ZO estimation would incur even higher overhead.
>
> **4. Weakness iv.** We thank the reviewer for bringing attention to this point and would like to **kindly remind the reviewer** that **our contributions extend beyond the general use of surrogate-based attacks.** **As stated** in the Introduction (Page 1, Lines 69–75), manipulating DL-based weather forecasts to intentionally alter downstream TC trajectories remains largely unexplored. Our method introduces domain-specific innovations required for this setting:  (a) adversarial optimization under a non-differentiable rule-based TC detector, supported by surrogate modeling (Page 3, Lines 111–119);  (b) a skewness-aware loss with kernel dilation to address the extreme sparsity of TC grids (<0.01%), which is not considered in prior adversarial literature (Page 3, Lines 133–146);  (c) distance-based perturbation control to maintain realistic and smooth TC trajectories without zigzag artifacts (Page 1, Lines 74–75). **These elements are not incremental reuse of generic techniques; rather, they are jointly necessary** to make targeted downstream trajectory manipulation feasible and physically meaningful. The improvements in Table 1 and Figure 3 confirm that conventional surrogate attacks cannot achieve this performance in the meteorological context.
>
> **5. Weakness v.** The generalization to other TC detection systems can be limited because different software may use different input variables, spatial resolutions, and physical rules, which naturally affect transferability. Our goal in this study is to demonstrate the feasibility of a representative operational detector. Evaluating additional detectors under varying configurations is a valuable extension that we identify as future work.

---

### Meta-Review · Area_Chair_B522 · 2025-12-23

**Summary:**

This paper studies adversarial attacks against weather forecasting models

Strengths:
(1) interesting and important research topic, (2) easy-to-following writing, (3) interesting integration between time series forecasting and tropical cyclone (TC) detection systems that addresses several critical challenges.

Weaknesses:
(1) marginal and incremental technical contribution, (2) vague motivation for surrogate model–based transfer attack, (3) inadequate empirical evaluation with only one TC detection system, and (4) lack of physics-aware stealthiness measure which might limit its real-world impact.

**Reviewer Concerns:**

Most of the concerns/weaknesses were not adequately addressed.

**Reviewer Scores:**

It is unlikely that reviewers will change their scores.

---

### Decision · Program_Chairs · 2026-01-26

Reject